# Thalamo-cortical spiking model of incremental learning combining perception, context and NREM-sleep

**Bruno Golosio** [1,2], **Chiara De Luca** [3,4] *, **Cristiano Capone** [4], **Elena Pastorelli** [3,4], **Giovanni Stegel** [5], **Gianmarco Tiddia** [1,2], **Giulia De Bonis** [4], **Pier Stanislao Paolucci** [4]

**1** Dipartimento di Fisica, Università di Cagliari, Cagliari, Italy, **2** Istituto Nazionale di Fisica Nucleare (INFN), Sezione di Cagliari, Cagliari, Italy, **3** Ph.D. Program in Behavioural Neuroscience, "Sapienza" Università di Roma, Rome, Italy, **4** Istituto Nazionale di Fisica Nucleare (INFN), Sezione di Roma, Rome, Italy, **5** Dipartimento di Chimica e Farmacia, Università di Sassari, Sassari, Italy

ᴥ These authors contributed equally to this work.
\* Chiara.DeLuca@roma1.infn.it

## Abstract

The brain exhibits capabilities of fast incremental learning from few noisy examples, as well as the ability to associate similar memories in autonomously-created categories and to combine contextual hints with sensory perceptions. Together with sleep, these mechanisms are thought to be key components of many high-level cognitive functions. Yet, little is known about the underlying processes and the specific roles of different brain states. In this work, we exploited the combination of context and perception in a thalamo-cortical model based on a soft winner-take-all circuit of excitatory and inhibitory spiking neurons. After calibrating this model to express awake and deep-sleep states with features comparable with biological measures, we demonstrate the model capability of fast incremental learning from few examples, its resilience when proposed with noisy perceptions and contextual signals, and an improvement in visual classification after sleep due to induced synaptic homeostasis and association of similar memories.

## Author summary

We created a thalamo-cortical spiking model (ThaCo) with the purpose of demonstrating a link among two phenomena that we believe to be essential for the brain capability of efficient incremental learning from few examples in noisy environments. Grounded in two experimental observations—the first about the effects of deep-sleep on pre- and post-sleep firing rate distributions, the second about the combination of perceptual and contextual information in pyramidal neurons—our model joins these two ingredients. ThaCo alternates phases of incremental learning, classification and deep-sleep. Memories of handwritten digit examples are learned through thalamo-cortical and cortico-cortical plastic synapses. In absence of noise, the combination of contextual information with perception enables fast incremental learning. Deep-sleep becomes crucial when noisy inputs are considered. We observed in ThaCo both homeostatic and associative processes: deep-sleep

**Data Availability Statement:** The model and data used to draw the results outlined in this manuscript are available at https://github.com/APE-group/ThaCo2.git and 10.5281/zenodo.4769175.

**Funding:** This work has been supported by the European Union Horizon 2020 Research and Innovation program under the FET Flagship Human Brain Project (grant agreement SGA3 n. 945539 and grant agreement SGA2 n. 785907; recipient Pier Stanislao Paolucci) and by the INFN APE Parallel/Distributed Computing laboratory. The funders had no role in study design, data collection and analysis, decision to publish, or preparation of the manuscript.

**Competing interests:** The authors have declared that no competing interests exist.

fights noise in perceptual and internal knowledge and it supports the categorical association of examples belonging to the same digit class, through reinforcement of class-specific cortico-cortical synapses. The distributions of pre-sleep and post-sleep firing rates during classification change in a manner similar to those of experimental observation. These changes promote energetic efficiency during recall of memories, better representation of individual memories and categories and higher classification performances.

## 1 Introduction

Increasing experimental evidence is mounting for both the role played by the combination of bottom-up (perceptual) and top-down/lateral (contextual) signals [1] and for the beneficial effects of sleep as key components of many high-level cognitive functions in the brain. In the following, we give an overview of some aspects, driven from experimental observations, that we have taken as fundamental building blocks for the construction of the model we present.

It is known that the cortex follows a hierarchical structure [2]; starting from this, Larkum et al. [1] propose an associative mechanism built-in at a cellular level into the pyramidal neuron (see Fig 1B), exploiting the cortical architectural organization (see Fig 1C). Long-range connectivity in the cortex follows the basic rule that sensory input (i.e., the feed-forward stream) terminates in the middle cortical layers, whereas information from other parts of the cortex (i.e., the feedback stream) mainly projects to the outer layers. This also applies to projections from the thalamus, a structure that serves as both a gateway for feed-forward sensory information to the cortex and a hub for feedback interactions between cortical regions. Indeed, only 10% of the synaptic feedback inputs to the apical tuft come from nearby neurons, and the missing 90% arise from long-range feedback connections. This feedback information stream is vitally important for cognition and conscious perception: this picture leads to the suggestion that the cortex operates via an interaction between feed-forward and feedback information. Larkum et al. [1] highlight that, counter-intuitively, distal feedback input to the tuft dendrite could dominate the input/output function of the cell: short high-frequency bursts would be produced on a combination of distal and basal input. As a consequence, although small (under-threshold) signals contribute only to their respective spike initiation zones, the fact that input has reached the threshold in one zone is quickly signalled to other zones. This provides the possibility for a contextual prediction: the activity in the apical tuft of the cell can lower the activity threshold driven by the basal region, the target of the specific nuclei in the thalamus that projects there the perceptual and feed-forward streams. In summary, this mechanism is ideally suited to associating feed-forward and feedback cortical pathways. Thus, they propose a conceptual interpretation of these biological pieces of evidence: the feedback signal aims at predicting whether a particular pyramidal neuron could or should be firing. Moreover, any neuron can fire only if it receives enough feed-forward input. Resulting from this interpretation, the internal representation of the world by the brain can be matched at every level with ongoing external evidence via a cellular mechanism, allowing the cortex to perform the same operation with massively parallel processing power.

Soft Winner-Take-All (WTA) plays an important role in many high-level cognitive functions such as decision making [3–5] classification and pattern recognition [6, 7]. Under a rough simplification, this mechanism can be realized through the competition among groups of excitatory neurons connected towards the same population of inhibitory neurons, which in turn is connected towards the excitatory groups it arbitrates [8–11]. Within appropriate conditions, the inhibitory signal will be sufficiently high to suppress the signal of all the low-firing

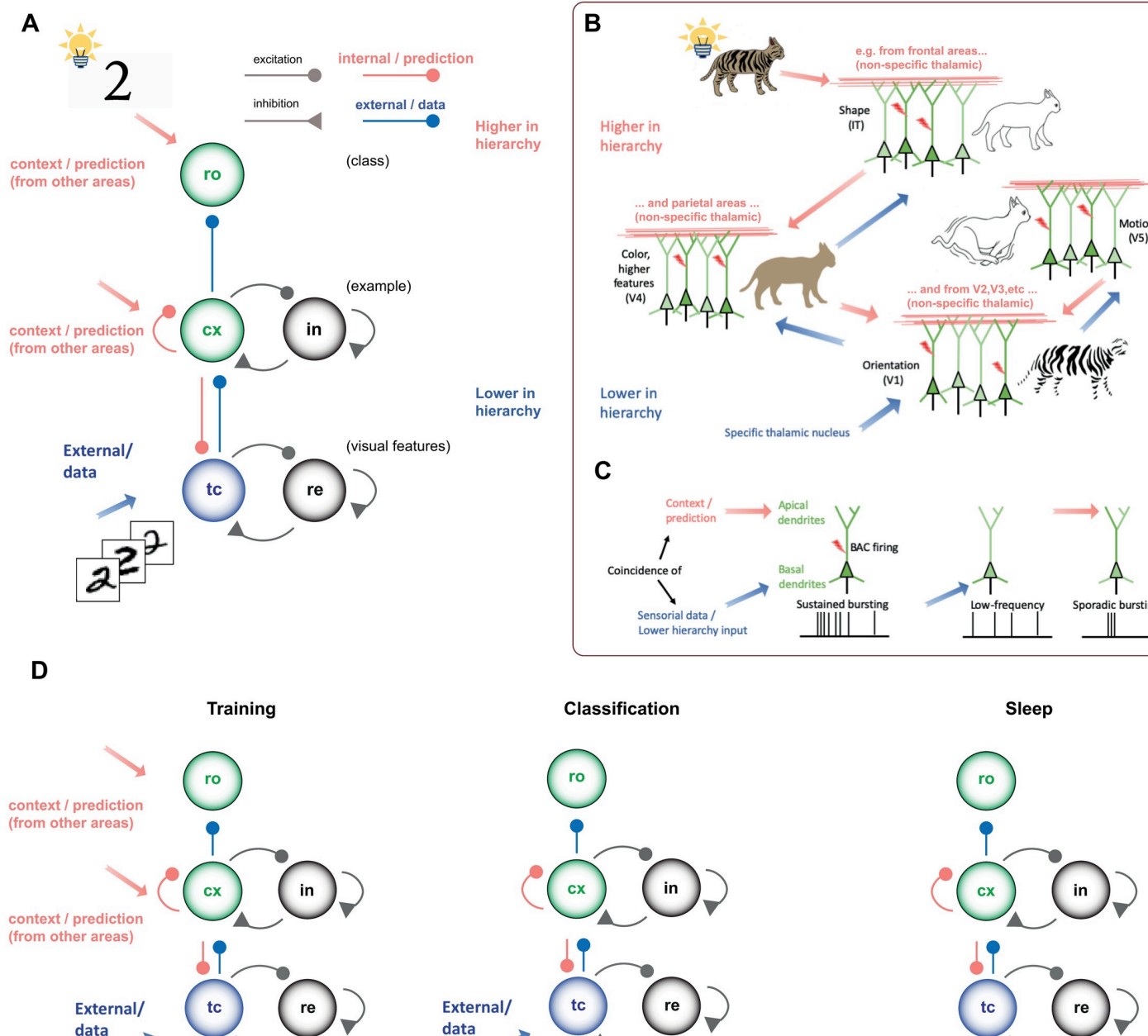

**Fig 1. Thalamo-cortical spiking model (ThaCo).** Panels B and C to compare the architecture of our model with the biological principles described in [1]. **A)** Scheme of the Thalamo-cortical spiking model (ThaCo). Input images, passed through a filter (HOG) are projected (blue arrow) to thalamic excitatory neurons (*tc*), mimicking the mechanism of the retinal *visual stimulus*. Thalamic neurons stimulate cortical excitatory neurons (*cx*) with a *perceptual feedforward* excitation (blue). Cortico-cortical and cortico-thalamic are considered as *top-down prediction* connections (red). (Red arrows—context/prediction) Currents coding for higher abstraction features incoming from other cortical areas. Cortical inhibitory neurons (*in*) arbitrate competition among cortical groups in a soft Winner-Take-All mechanism (WTA). Inhibitory reticular neurons (*re*) control the thalamic firing rate. The cortical layer is in turn connected to readout neurons (*ro*) **B)** A cellular mechanism for associating feed-forward and feedback signals. Low-level features are encoded in primary sensory regions and this signal propagates up the visual hierarchy (e.g. striate cortex (V1) sensitive to orientation, V4 sensitive to colour, V5 sensitive to motion, and inferior temporal (IT) cortex sensitive to shapes and objects). Higher-level areas provide feedback information (context or expectation) to lower areas. The ThaCo model presented in this paper is a single area model and the contextual signal is assumed to collect during training the knowledge carried by all other areas in the hierarchy (see red arrows in panel A). **C)** Conceptual representation of the back-propagation activated calcium (BAC) firing hypothesis supporting efficient binding of features and recognition. Pyramidal neurons receiving predominantly feed-forward information are likely to fire steadily at low rates, whereas the simultaneous presence of contextual and perceptual streams changes the mode of firing to bursts (BAC firing). This coincidence mechanism is mimicked in our ThaCo model. **D)** During training (left), the injection of *contextual signal*, plays the role of internal prediction and increases the perceptual threshold of a subset of cortical neurons. The simultaneous presence of *perceptual* and *contextual* promotes a high firing rate in such neurons, mimicking the BAC mechanism. Also, the simulataneous presence of the signal from the cortical layer and of the contextual signal promotes a high

firing rate in readout neurons. In the classification phase (centre) the contextual signal is turned off. In the sleeping phase (right), the sensory pathways are turned off, and all the activity is generated spontaneously.

excitatory groups of neurons whereas the high-firing ones survive. Such conditions can be achieved thanks to synaptic plasticity, which strengthens the connections among neurons of the same group and weakens those among competing groups, coupled with homeostatic mechanisms [12, 13].

Spike-timing-dependent plasticity (STDP) has been proposed as one of the essential learning ingredients in the cortex [14–18]. According to this plasticity rule, if the postsynaptic neuron fires an action potential just after a presynaptic spike, the synaptic weight will increase, whereas in the opposite case it will decrease. Through this mechanism, the synapses connecting neurons correlated by a principle of causality are metabolically rewarded. Chen et al. [19] have shown that networks of excitatory and inhibitory spiking neurons with either STDP or short-term plasticity can generate dynamically-stable WTA behaviour under certain conditions on initial synaptic weights.

Another key aspect that we consider in this study is the role of sleep during learning. Sleep is essential in all animal species, and it is believed to play a crucial role in memory consolidation [20, 21], in the creation of novel associations, as well as in the preparation of tasks expected during the next awake periods. Indeed, young humans pass the majority of time sleeping, and the youngest are the subjects that have to learn at faster rates. In adults, sleep deprivation is detrimental for cognition [22] and it is one of the worst tortures that can be inflicted. Among the multiple effects of sleep on the brain and body, we focus here on the consolidation of learned information [23]. Homeostatic processes could normalize the representation of memories and optimize the energetic working point of the system by recalibrating synaptic weights [24] and firing rates [25]. Specifically, Watson et al. [25] show that fast-firing pyramidal neurons decrease their firing rates over sleep, whereas slow-firing neurons increase their rates, resulting in a narrower population firing rate distribution after sleep. Also, sleep should be able to select memories for association, promoting higher performance during the next awake phases [26]. Indeed, Capone et al. [27] demonstrate the beneficial effects of sleep-wake phases involving homeostatic and associative processes in a visual classification task. Indeed, in [27], some of us illustrated how to assemble a simplified thalamo-cortical spiking model that can both express deep-sleep-like oscillations (in the form of an emergent, self-induced network phenomenon) and enter an awake-like asynchronous regime. This dynamical behaviour has been obtained by changing a few parameters in the equation that describes the dynamics of excitatory neurons in the spiking model, and thus exploiting a well established modelling principle that represents a few prominent features of brain-state acetylcholine-mediated neuromodulation, able to induce in the model the transition between awake-like asynchronous and deep-sleep-like oscillatory regimes [28]. However, this neuromodulation modelling principle has neither been previously applied to the study of the deep-sleep cognitive effects nor to simulations of learning-sleep cycles (such as in our previous work [27] and in this study). Specifically, the spiking model we propose is trained on a set of training patterns (here, on images of handwritten digits) and then exposed to never-seen examples to be classified (here, among human-assigned digit classes). Also, the model structure proposed in [27] and adopted in this work, is able to perform an asynchronous awake-like state of the network, by acting on the neural dynamics parameters. When the prescribed changes in the neural parameters induce the network to express deep-sleep-like oscillations, STDP is observed to produce in the model the spontaneous emergence of a differential homeostatic process. First, a down-regulation emerges of the stronger synapses created by the STDP during the training,

those synapses that connect the best-tuned neurons during the training phase on each training example. At the same time, we observe that STDP increases the strength of synapses among neurons tuned on patterns belonging to the same class. Such hierarchical, spontaneous reorganization promotes better post-sleep classification performances. In short, the underlying mechanism is based on the similarity among thalamic coding of training examples belonging to the same class. During deep-sleep oscillations, such similarity supports the preferential activation of thalamo-cortico-thalamic connection paths among neural groups tuned to training examples belonging to the same class, and the consequent coactivation and unsupervised strengthening of class-specific synapses. This point has been illustrated in [27].

Combining the above-described set of cortical principles, we aimed at creating a simplified, yet biologically-plausible, thalamo-cortical spiking model (ThaCo, see Fig 1A). ThaCo exploits the combination of contextual and perceptual signals to construct a soft Winner-Take-All mechanism (WTA) capable of fast learning from few examples [29] in a synaptic matrix shaped by spike-timing-dependent plasticity (STDP). ThaCo has been calibrated to express deep-sleep-like activity and to induce modifications to the distributions of pre- and post-sleep firing rates comparable to biological measures like those carried out by Watson et al. [25] for an investigation of the deep-sleep effects on learning and classification (another beneficial aspect, the recovery and restoration of bio-chemical optimality, is not considered at this level of abstraction). In the context of machine learning, a distinction is made between instance-incremental methods, which learn from each training example as it arrives, and batch-incremental methods, in which the training data are organized in groups of examples, called batches, and the model is trained only on complete batches [30]. Depending on how different classes are represented by the examples in the batches, there are three training schemes [31]: new instances (NI), in which each new batch contains different instances of the same classes represented in previous batches, new classes (NC) in which examples belonging to novel classes become available in subsequent batches, and new instances and classes (NIC) in which subsequent training batches contain examples from both known and new classes. However, it should be noted that when it is necessary to constantly evaluate the performance of incremental learning, for reasons of computational efficiency the training set is divided into batches even for models capable of instance-incremental learning. Shimizu et al. [32] propose a training method based on balanced mini-batches, which reduces the effect of imbalanced data in supervised training. Our work is focused on instance incremental learning, and the training scheme is based on balanced mini-batches. Specifically, with ThaCo we investigated several brain aspects and learning capabilities: 1- incremental learning from few examples; 2- resilience to noise when trained over degraded-quality examples and asked to classify corrupted images; 3- comparison with the performances of knn algorithms; 4- the ability to fight noise in the contextual signal thanks to the introduction of a biologically-plausible deep-sleep-like state, inducing beneficial homeostatic and associative synaptic effects.

## 2 Results

In this work we test the capability of the implemented thalamo-cortical network model (ThaCo) of expressing incremental learning when trained to learn and recall images (from the MNIST dataset), and we investigated the role and the mechanisms of the occurrence of biological-like deep-sleep dynamics. First, we present a comparison of the ThaCo model behaviour with the biological observations made by Watson et al. [25] on the changes of firing rate distributions in awake, sleep and post-sleep phases (see Fig 2). Indeed, since one of the goals of this work is to implement a biologically-plausible model capable to display different "cognitive states", the comparison with experimental outcomes is important to question its plausibility.

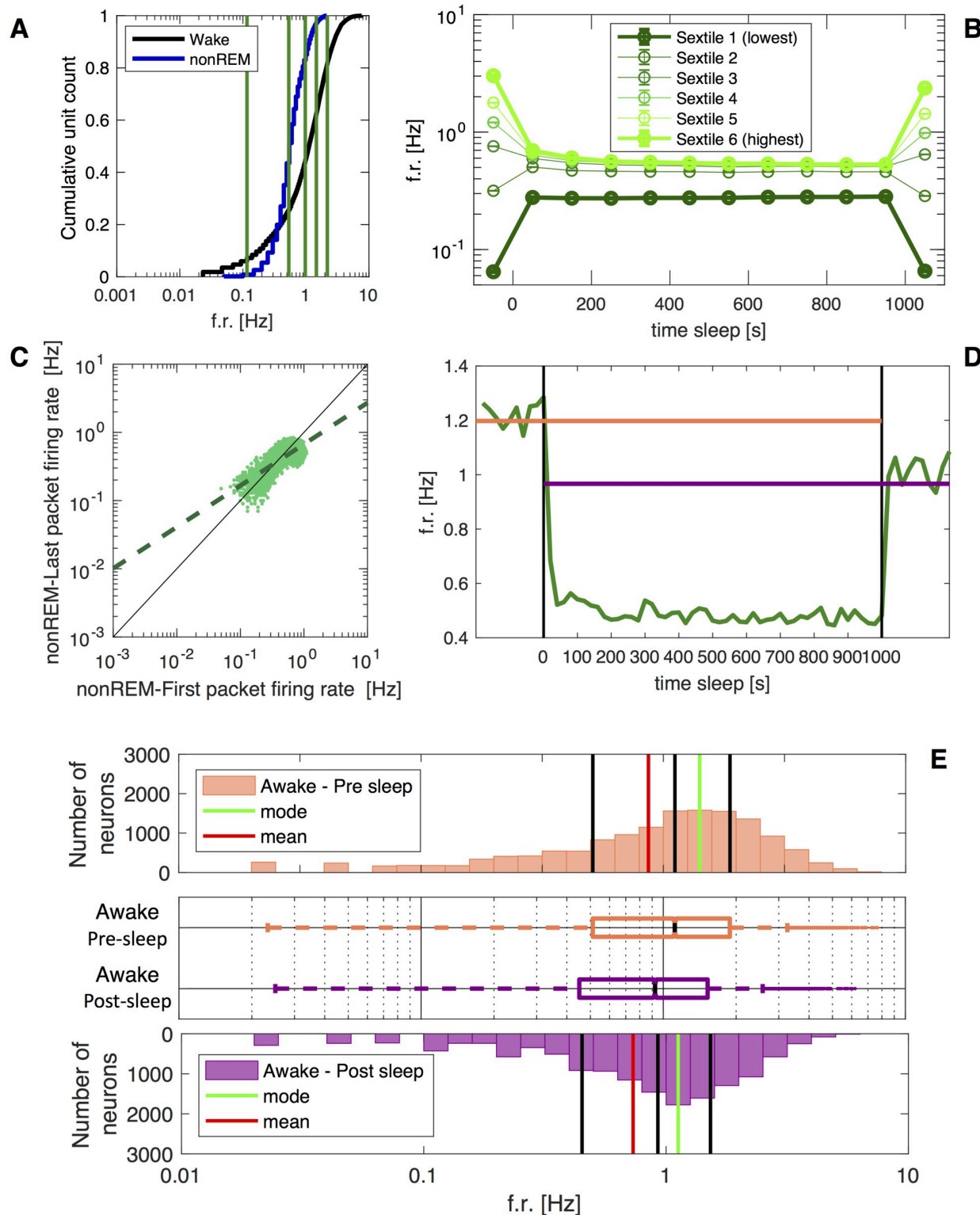

**Fig 2. Sleep-like features. A)** State-wise differences of average firing rate, to be compared with Fig 2A by Watson et al. [25]. Cumulative distribution of the firing rates of individual cortical neurons (log scale); note the brain-state dependent differences (colour). Vertical lines separate neurons sorted by AWAKE firing rates into six subgroups (sextiles) with an equal number of elements. **B)** Firing rate changes across sleep in each of the six groups defined by the awake firing rates, to be compared with Fig 3B by Watson et al. [25]. High firing rate neurons show decreasing activity; low firing rate neurons do not increase their activity over sleep. **C)** Opposite modulation of neurons of different firing rates, to be

compared with Fig 3D by Watson et al. in [25]. Comparison of individual neuron firing rates during the first and last packet of sleep. The regression line is significantly different from unity, showing that high and low firing rate neurons are oppositely modulated over sleep. **D)** Cortical neuron population mean firing rate changes across sleep, to be compared with Fig 3B by Watson et al. in [25]. **E)** Awake firing rate distribution of cortical neurons pre-sleep (upper plot) and plot-sleep (lower plot). Solid lines depict descriptive statistics parameters: Q1, 25% quartile; Q2, 50% quartile (median); Q3, 75% quartile. Middle plot: boxplots of the distributions. The central mark indicates the median, and the bottom and top edges of the box indicate the 25th and 75th percentiles, respectively.

After the validation obtained against experimental results, we demonstrate the capability of the model to learn incrementally, i.e. to continuously extend its knowledge by learning from new training examples while retaining most of the previously acquired memories. The learning ability of the model was assessed using an approach that alternates incremental training with tests meant to evaluate the pre-sleep and post-sleep classification performance. During the training phase, samples are randomly extracted from the training set of the MNIST database; they are given in input to the system together with example-specific contextual signals that reach the cortical neurons, and a digit-class-specific contextual signal that reaches only the read-out neurons. Notably, to stress out the difference with respect to the training phase, during the classification phase no contextual signal is transmitted: the response of the network is recorded, based on the firing rates of the excitatory neurons of the cortex. It should be emphasized that the proposed model is not an engineering solution to the problem of incremental learning in pattern classification, but a simplified model of the low-level processes that supports and emulates the ability to learn incrementally in the biological brain. Indeed, the size of the training set is relatively small compared to those often used in machine learning, and some issues that are of primary importance for both artificial and biological incremental learning, such as catastrophic forgetting, go beyond the aims of this paper. Then, We measure the model incremental classification performance and we compare it to that expressed by the K-Nearest neighbour family of artificial incremental learning algorithms (specifically Knn-1, Knn-3, Knn-5). Knn is an extensively used classification algorithm, which has been succesfully applied to a wide range of problems in different fields. Furthermore, unlike many other classification systems used in machine learning, the Knn family is suitable for incremental learning and also it works relatively well even with few training examples and, for large enough training sets, the Knn algorithm is guaranteed to yield an error rate no worse than twice the Bayes error rate, which is the minimum achievable given the distribution of the data [33]. For these reasons, the Knn classifier has been chosen as reference for the evaluation of the classification ability of the proposed system. We show that—even without the beneficial contribution of sleep—this model shows higher resilience to noisy inputs than Knn. Finally, we demonstrate the beneficial effects of deep-sleep-like cortical slow oscillations on the post-sleep classification accuracy of MNIST characters when a noisy contextual signal is injected during the awake training (a situation that could be interpreted both as the case of different levels of prior knowledge about the correct classification label of the current example during the training, and as related to the largely stochastic nature of cortical organisation and of the activity of other cortical areas).

## 2.1 ThaCo model pre- and post-sleep firing rates and comparison with the experiments

We compare the network behaviour of the ThaCo model during three simulated phases (pre-sleep awake-like, deep-sleep-like and post-sleep awake-like, see Fig 2) with those observed in rats by Watson et al. in [25]. When approaching the design of the ThaCo spiking model, an improvement of what some of us presented in [27], we relied on the well-established framework of Mean-Field theories [34], [35] to construct a network capable of spontaneously

displaying two different dynamical regimes. This is obtained by acting on some parameters of the excitatory neurons (specifically, the spike frequency adaptation (SFA) and the excitatory synaptic conductance), to model acetylcholine-mediated neuromodulation on neural dynamics that supports the transitions between awake-like asynchronous activity and deep-sleep-like slow oscillations [28]. Specifically, in Fig 3A we show the incoming current to each cortical neuron versus its adaptation current during different network stages (pre- and post-sleep classification, beginning and end of the sleeping phase). In particular, sleep states are characterized by high levels of spike frequency adaptation currents (obtained through a modulation of the SFA parameter), inducing oscillations. Moreover, late sleep and after-sleep classification have low levels of input currents, due to a sleep-mediated synaptic depression leading to a reduction in the current circulating in the network. When set in the deep-sleep state, a non-specific stimulus, administered at a low steady firing rate to cortical neurons, is sufficient to elicit the emergence of cortically-generated Up-states and of thalamo-cortical Slow Oscillations (SO). As shown in top lines of Fig 3B and 3C, in the SO regime, the thalamo-cortical spiking network displays a firing rate oscillation frequency between $0.25Hz$ and $1.0Hz$ and durations of Up-states (a few hundreds of ms) comparable with experimental observations in deep-sleep recordings. During the initial stages of SO, Up-states are independently sustained by neuron populations tuned to specific images memorized during the training phase, and tend to reactivate thalamic neuron coding for the memorized images. Then, thanks to the similarity among training instances, the recruitment of other neural groups in the cortex is promoted. This creates preferential cortico-talamo-cortical excitatory pathways, inducing an STDP-mediated association of cortical neurons previously tuned to training instances that expressed similar thalamic representation (see Fig 3B and 3C). We name *top-down prediction* such cortico-thalamic activation that spontaneously occurs during SO. During the sleep period, thanks to cortico-cortical plasticity, the coactivation of neurons originally tuned to training instances of the same class becomes a typical feature of each Up-state: the WTA mechanism cooperates in selecting different neuron codings for different classes during each Up-State. Another key aspect is the generalized homeostatic depression, which is known to happen during deep-sleep and serves as a protection, to prevent Up-state-mediated associations that could drive towards a fully associated network. This effect is modelled thanks to the Non-Linear Temporal Asymmetric Hebbian (NLTAH) learning rule of the STDP we used [36], which reduces the strength among the most frequently coactivated neurons, leading to a progressive reduction of mean firing rates and frequency of the Up-States (see Fig 3B and 3C, top rows) which is consistent with experimental observations, in particular for what concerns the decrease of SO frequency during the night course [37]. The first noteworthy result presented in this paper is that this new calibration of the model greatly enhances the match with experimental data, as detailed in the following. Indeed, while the model [27] was already able to express the transition between states [38] such as sleep-like slow oscillations activity and awake-like classification, the refinements of its parameters here introduced make ThaCo more biologically plausible, leveraging as calibration tool the accurate comparison with experimental observations of differential changes in firing rates. In their work, Watson et al. [25] used large-scale recordings to examine the activity of neurons in the frontal cortex of rats and observe the distributions of pyramidal cell firing rates in different brain states: Awake, REM, nonREM and Microarousals. They found that periods of nonREM sleep reduced the post-sleep awake activity of neurons with high pre-sleep firing rate while up-regulating the firing of slow-firing neurons. Moreover, in their experiments, the neuronal firing rate varied with the brain state and, across all states, the distribution of per-cell mean firing rates was strongly positively-skewed, with a lognormal tail towards higher frequencies and a supra-lognormal tail towards lower frequencies. We set out the model parameters to reproduce these measures. In Fig 2A we present the cumulative

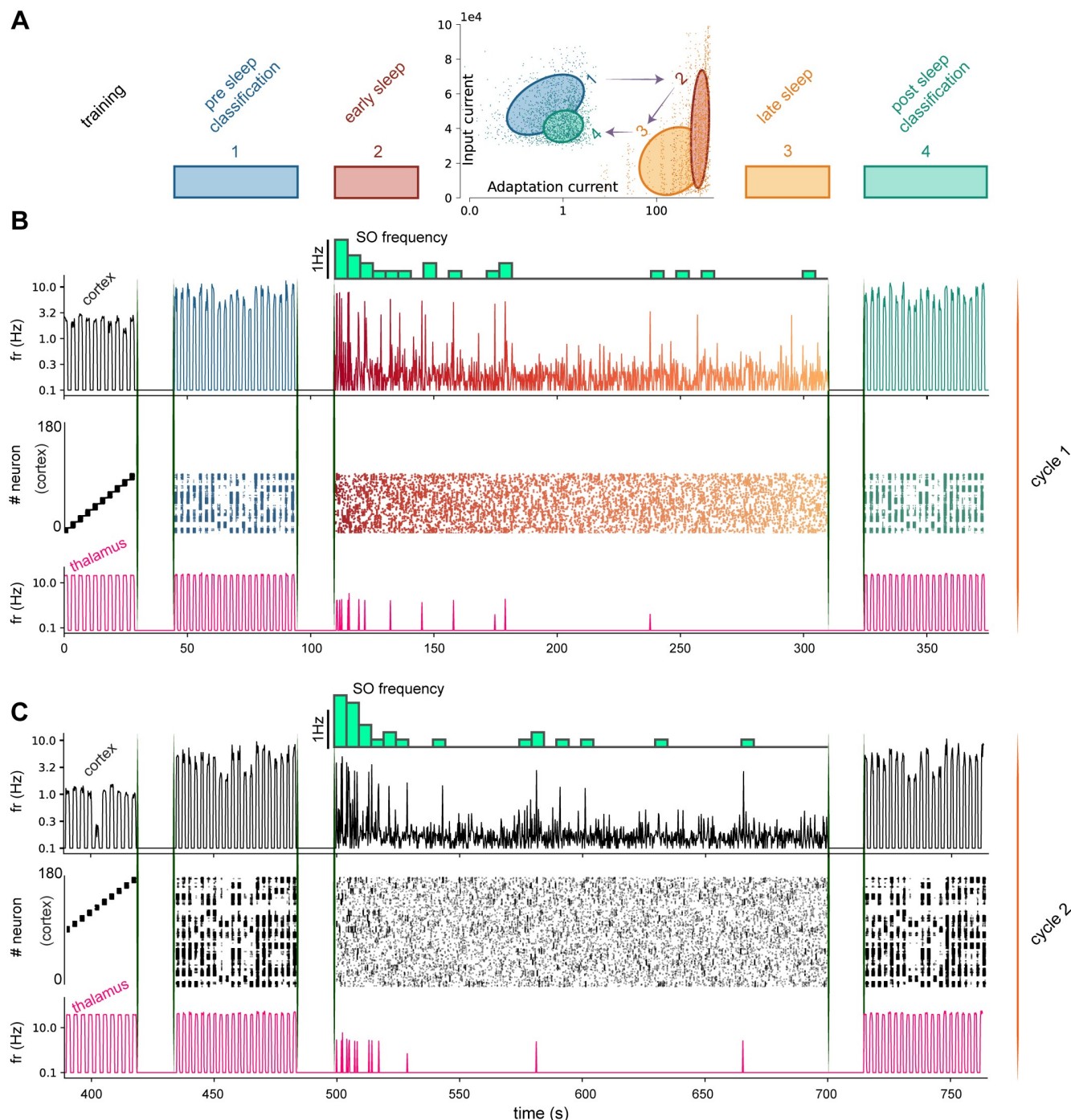

**Fig 3. Incremental learning with alternation of awake training and sleep in ThaCo. A)** *Dots*: incoming current to each cortical neuron versus its adaptation current during different network stages of the activity presented in B. Pre-sleep classification phase (represented in blue, number 1), early sleep-like phase (in red, number 2), late sleep-like phase (in orange, number 3) and post-sleep classification phase (in green, number 4). *Ellipses*: Areas in the plot associated with different network stages, estimated from data through a Gaussian Mixture Model with a full covariance matrix. During sleep, the total input current to the cortical neurons decreases due to the sleep-induced homeostatic effect, that reduces recurrent connections weights in the cortical layer (see the transition from number 2 to number 3 in the diagram), notwithstanding the constant external aspecific stimulus. Sleep-like activity, on the other hand, affects the network status during the following awake classification phase 4: the effect of STDP during sleep is a general reduction and homogenization of input current distribution, as shown in a comparison between the pre-sleep stage 1 and post-sleep stage 4 in the diagram. **B** and **C)** Spiking cortical and thalamic activity produced during training (10 examples, one per digit class), classification (20 images) and sleeping phase for two consecutive sets respectively. First row: mean firing rate of the cortical neurons trained over a set of 10 examples (each set is used to independently train 200 cortical neurons, 20 per digit example); during the sleeping phase, the slow oscillation frequency trend in time is also depicted. Second row: raster plot of the first 400 cortical neurons. Third row: mean firing rates of thalamic neurons. Once recruited in the training phases, the cortical neurons participate in classification and sleeping phases.

distribution of neuronal mean firing rates for both awake and nonREM states of our model, to be compared with Fig 2A of Watson et al. [25] (REM not included in ThaCo). Median rates (± SD) of excitatory cortical neurons in ThaCo in each state are: awake, $1.2 \pm 1.1 Hz$ and nonREM, $0.6 \pm 0.3 Hz$.

An interesting feature is that lognormal distributions spontaneously emerge from our simulations. This result is coherent with experimental observations and with theoretical considerations showing that the lognormal distribution of activities in randomly connected recurrent networks is a natural consequence of the non-linearity of the input-output gain function [39]. In agreement with Watson et al. [25], we also found that the arithmetic mean of the population firing rates declined throughout sleep, as visible using a test of correlation of spike rate versus time (see Fig 2D to be compared with Fig 3B by [25], the slope of the rate change within time-normalized sleep from all $cx$ neurons in all recordings is $R = -0.10$, $p = 10^{-3}$). In order to demonstrate that sleep brings varying differential effects across the rate spectrum, we compared mean firing rates in the first and the last $100s$ of sleep. As depicted in Fig 2C, fast-firing neurons decreased their rates over sleep, whereas slow-firing neurons increased their rates (to be compared with Fig 3D by [25]). To quantify this observation, we assessed spike rates of the same neurons in the first versus the last nonREM $100s$ of sleep and found the slope of this correlation significantly departed from unity (slope, 95% confidence interval $0.6015 - 0.6130$).

Furthermore, following [25], we divided ThaCo excitatory cortical neurons into six sextile groups sorted by their awake firing rates (Fig 2A). As shown in Fig 2B, the sextile with the highest firing rates significantly decreased its activity over sleep, in accordance with results obtained by Watson et al. [25] (see Fig 3B of their work). Finally, we evaluated the impact of sleep on the cortical firing rates distribution during awake states. In Fig 2E, we compare firing rates distribution pre- and post-sleep depicting the homeostatization effect of sleep.

## 2.2 The ThaCo network model and the training protocol

The proposed ThaCo circuit is organized into three layers, as shown in Fig 1A: an input layer, the *thalamus*, which consists of an excitatory population (*tc*) whose firing rate is under the control of a *reticular* inhibitory fully-connected population (*re*); the *cortex*, consisting of an excitatory population (*cx*) and an inhibitory population (*in*), both fully connected as well; a *readout* (*ro*) layer, to which the cortex is also fully connected, composed of subgroups of neurons of neurons associated to each class. The learning protocol is organized in alternation of training phases—when the internal structure of the network is shaped according to the learnt examples—and testing phases—when the classification performance of the network is evaluated (see Section 4.1). In both training and classification phases, the network is provided with sample images drawn from the MNIST dataset. The sample images are pre-processed to produce stimulus signals that are transmitted to the excitatory neurons of the thalamus (see paragraphs "The datasets of handwritten characters" and "Thalamic coding of visual stimuli" in S1 Text for more details). During the training phase, simultaneously with the input sensory-like stimulus, contextual signals are transmitted to the excitatory neurons of the cortex and to the readout neurons. The observed bursting behaviour of the neurons is a consequence of the temporal coincidence between impinging perceptual and contextual signals. Specifically, for each example to learn, an example-specific group of excitatory neurons in *cx* is facilitated through the presentation of a contextual signal. This induces a higher activity in these neurons causing a strengthening of both thalamo-cortical synapses and recurrent synapses. This example-specific tuning involves each neuron with a single training example only, whose category defines (in an unsupervised manner) a natural category for which the neuron is better tuned. Meanwhile, a subgroup of readout neurons (*ro*) is stimulated by a digit-class specific contextual

signal, leading to an enhancement of connections between the cortical neurons trained over the presented example and the subgroup of readout neurons associated with the correct class. The simultaneous stimulation by perceptual signals and contextual signals emulates the organizing principle of the cerebral cortex as described by Larkum et al. [1], approximating the effects of the dendritic apical amplification mechanism at the cellular level. It is worth noting that this is the only phase when a category-specific (rather than an example-specific) signal is given to the network: protocols concerning this *ro* layer are *supervised* training protocols, whereas those for the other layers can be referred as *unsupervised* training protocols. During the classification phase, signals resulting from preprocessed images are again transmitted to the thalamus analogously to the training phase, however, no contextual signal is transmitted to either cortical and readout neurons. During this stage, the neuronal activation results from the combination of the current injected by perceptual signals and the one injected by recurrent interconnections strengthened by the synaptic STDP dynamics during the training and modified by STDP during sleep cycles. We infer the network answer to the classification task in two different ways: first, unsupervised, taking the class of the example over which the most active subgroup of cortical neurons has been trained; second, supervised, taking the class associated to the most active subgroup of readout neurons. Specifically, the readout layer performs the integration of signals coming from the subgroups of cortical neurons trained over different examples belonging to the same class (see Section 4.1 for a more detailed representation of the learning process). The activity produced by the cortical neurons during training, classification and sleeping phases is depicted in Fig 3B and 3C.

We set the network parameters in an *under threshold* regime that enables the training above described through the selected STDP model on the single-compartment standard Adaptive Exponential (AdEx) integrate-and-fire neuron that would not otherwise distinguish among basal and apical stimuli. See details about the model construction, the presentation of visual stimuli and the addition of noise in the Material and Methods, Section 4 and in S1 Text.

## 2.3 Incremental learning: Performances

We trained the proposed network over an incremental number of training examples and evaluated its classification performances on a set of images never shown. We also compared the average accuracy of our thalamo-cortical spiking model with that obtained using standard Knn-x classification systems for different numbers of training examples per digit category. See Fig 4 and Table 1. The model presented in this work enables instance-incremental as well as class-incremental learning. The training protocol we adopted for the results presented here was based on the balanced-mini-batches scheme proposed by [32]. More specifically, the training set of hand-written digits was divided into mini-batches of 10 examples each, in which each class was represented by just one example. In S1 Text, we include a comparison of performance obtained using different training protocols.

MNIST images have been presented to the ThaCo *th* layer using the improved pre-processing protocol described in paragraph Thalamic coding of visual stimuli of the S1 Text. The accuracy has been evaluated over classification trials, each one including 500 images, and the classification accuracy has been averaged over 20 trials. Fig 4A shows the accuracy for incremental learning as a function of the number of training examples per class. Fig 4B and 4C, on the other hand, depict the average accuracy of the compared training algorithms for the last 10 to 20 and the first 1 to 5 training examples per class respectively, to better show their different behaviour at different stages of the learning process. For the MNIST dataset, higher-order Knn algorithms surpass the performance of Knn-1 only when the training set includes more than 10 examples per digit class. It is worth noting that the soft WTA mechanism of ThaCo can

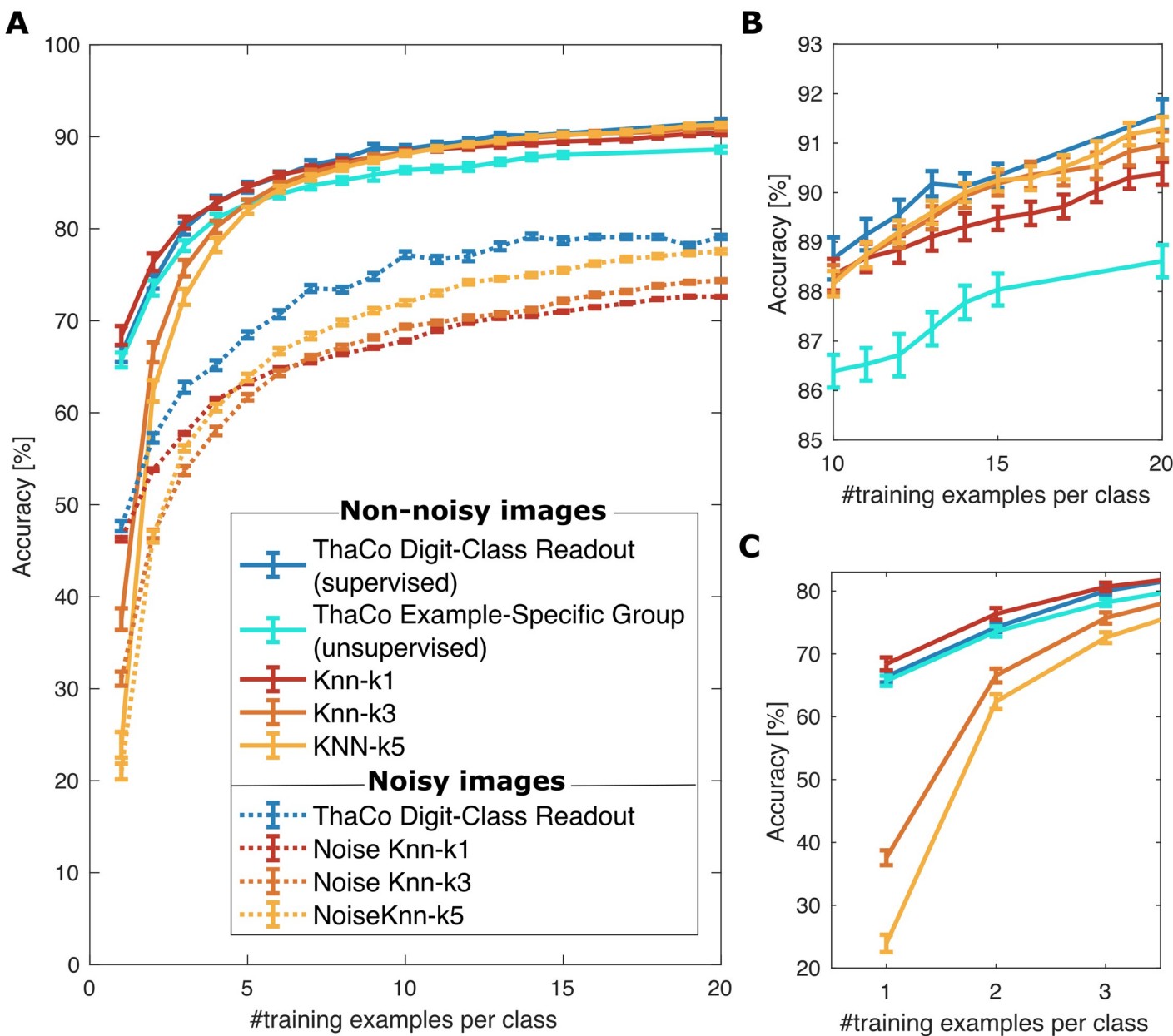

**Fig 4. Comparison of the average accuracy of the proposed thalamo-cortical spiking model compared to artificial K-nearest-neighbour incremental algorithms in absence of noise (solid lines) and with noisy inputs(dotted lines). A)** We infer the network answer to the classification task in two different ways: 1) Digit-Class Readout, as the class associated to the most active subgroup of readout neurons in ThaCo (supervised approach); 2) Example-Specific Group, by mapping the class over which the most active subgroup of cortical neurons has been trained (unsupervised approach). Solid lines depict accuracy in absence of noise, dotted lines depict accuracy with *"Salt and Pepper"* noise (density = 0.2) injected into the unprocessed MNIST images for both training and classification phases. Accuracy is assessed on an independent test set, consisting of 500 examples; the average and standard error of the mean (SEM) are evaluated on 20 different trials using independent training sets in each. **B), C)** represent the same plots shown in A) on different scales, for visualization purposes and for highlighting selected features. Specifically, **B)** ThaCo behaves like Knn-k1 for a small number of examples; **C)** as the number of training examples increases, ThaCo Digit-Class-Readout exhibits performances that are comparable with higher-order Knn algorithms.

learn incrementally and has comparable performances to the best Knn-n algorithm for a given number of training examples. Specifically, the supervised ThaCo is proven to be able to perform the integration of signals coming from subgroups of cortical neurons trained over different examples belonging to the same class and its performances are comparable to higher-order

**Table 1. Accuracy achieved by the different learning algorithms over a different number of training examples.** The accuracy has been computed over a test set of 500 examples, and the average is done over 20 trials.

| | Accuracy (%) | | | | | |
|---|---|---|---|---|---|---|
| | Training examples per class | | | | | |
| Algorithm | 1 | 2 | 3 | 5 | 10 | 20 |
| Knn, k = 1 | **68.0 ± 1.0** | **76.1 ± 1.0** | **80.6 ± 0.8** | **84.5 ± 0.4** | **88.3 ± 0.4** | 90.4 ± 0.3 |
| Knn, k = 3 | 37.2 ± 1.3 | 66.0 ± 1.1 | 75.4 ± 0.9 | 82.7 ± 0.4 | **88.3 ± 0.3** | **91.0 ± 0.3** |
| Knn, k = 5 | 23.9 ± 1.5 | 62.0 ± 1.3 | 72.2 ± 0.9 | 81.9 ± 0.4 | 88.2 ± 0.3 | **91.2 ± 0.3** |
| ThaCo—Digit class readout | **65.7 ± 1.0** | **74.8 ± 0.8** | **80.4 ± 0.7** | **84.8 ± 0.6** | **88.6 ± 0.5** | **91.1 ± 0.3** |
| ThaCo—Example specific group | **65.7 ± 0.9** | **73.6 ± 0.9** | 78.1 ± 0.6 | 82.7 ± 0.4 | 86.4 ± 0.4 | 88.6 ± 0.4 |

Knns, whereas the unsupervised ThaCo performances are proven to be comparable with Knn-1 performances when few examples are presented.

## 2.4 Classification of noisy input

We evaluated the network behaviour within a noisy input environment and compared it to the Knn performances. For this, we injected a *'Salt and Pepper'* noise [40] (density = 0.2) into the unprocessed MNIST images. Noisy images are then pre-processed (see 1) and presented to the network in both training and classification phases. Fig 4A depicts the average accuracy of the network trained incrementally over a total number of 20 noisy examples per class and compares it to performances of Knn-n algorithms (as in section 2.3, both Knn and ThaCo algorithms are trained incrementally). It is worth noting that in this scenario the ThaCo algorithm has better performances than the Knn-n algorithms.

## 2.5 Beneficial effect of deep-sleep in compensating the impact of noisy contextual labels

For the aim of introducing more biologically-plausible elements regarding the combination of contextual and perceptual signals, we slightly modify the ThaCo training protocol: the magnitude of the contextual signal given to both the cortex and the readout layer trained over a new learning example is now randomly extracted from a Gaussian distribution. As a consequence, some of the presented examples are better represented than others, resembling a more realistic situation in the cortex in which both the degree of knowledge projected by other areas and the number and strength of apical synapses carrying the contextual information and raising the perceptual thresholds during learning are not exactly equal for all the presented examples and all the neurons in the selected group.

We introduce the deep-sleep state in our training protocol, as follows: after each training phase, we disconnect ThaCo from external inputs and induce deep-sleep-like oscillations, following the method described in [27] and here described in Section 2.1 and in S1 Text, paragraph Sleep-like oscillatory dynamics. As expected, noise in the contextual signal leads to a drop in performance, compared to the idealized situation presented in the previous section (*i.e.* the careful equalization of contextual signal), but such drop can be reduced by sleep, as shown in Fig 5A.

At the synaptic level, it is possible to observe how deep-sleep-like slow oscillations induce in the current ThaCo model both a regularisation of the strength of the memories of individual learned examples through homeostasis and an association between groups of neurons trained over different examples of the same class. Figs 5B and 6 report such sleep-induced optimization of the synaptic representation of memories. Specifically, within neurons belonging to the

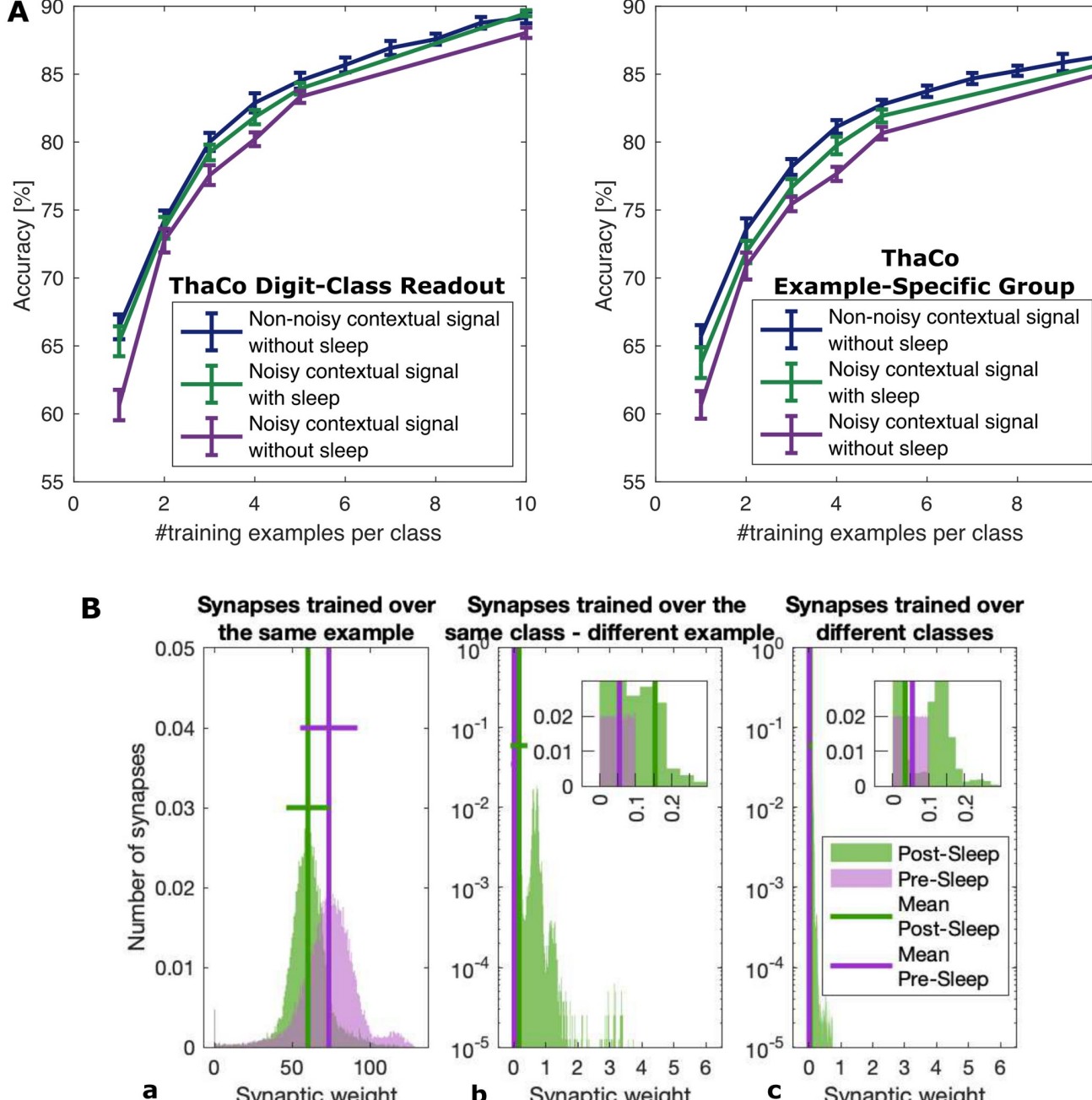

**Fig 5. A) Sleep mitigates the effects of noisy contextual signals on the classification performances of ThaCo** Classification performances evaluated over the Readout layer (supervised learning protocol, left) and Cortical layer (unsupervised learning protocol, right). Comparison among the network trained over non noisy examples (blue), the network trained over noisy examples without any deep-sleep like phase (violet), and the network trained over noisy examples interposing a deep-sleep-like activity between the training and the classification phases (green). The contextual signal provided in the training phase is corrupted by noise (i.e. some examples are associated with stronger synapses), leading to a drop in performances (comparison between blue and violet line). Still, the interposition deep-sleep-like phases between noisy-training and classification phases recovers the performances of the network trained with a non-noisy protocol. **B) Sleep-induced homeostatic and associative effects on cortico-cortical synaptic-weight distributions**. Pre-sleep (violet), post-sleep (green). Solid lines: mean and standard deviation. **a)** Intra-group connections: weight distributions of synapses connecting neurons trained over the same example (i.e. that during the training stage were triggered by the same contextual stimulus, thus activated simultaneously during a specific training example); **b)** Intra-class connections: weight distributions of synapses connecting neurons trained over different examples belonging to the same class (i.e. that have not been simultaneously triggered by the contextual stimulus in the training phase, but still have been triggered by a sensorial thalamic signal associated to images belonging to the same class) **c)** Inter-class connections: Connections among groups trained over different classes (i.e. triggered by the contextual stimulus together with a sensorial thalamic signal associated to images belonging to different classes). We note the homeostatic effect of sleep (in A) leading to a general reduction of weights associated to example-specific synapses and a reinforcement of the intra-class connections (in B). Synapses

connecting groups trained on different examples, on the other hand, are much less affected by sleep. The inset, showing part of the same plot in a lin-lin scale, is added to illustrate the shape of the pre-sleep distribution, and the difference in the mean values before and after sleep.

same example-specific group, the synaptic weights distribution decreases its mean and coefficient of variation (from $\mu = 74$, $\mu/\sigma = 0.3$, skewness 0.55 pre-sleep to $\mu = 60$, $\mu/\sigma = 0.2$, skewness $-0.26$ post-sleep) whereas within neurons belonging to different example-specific group but coding for the same class, the synaptic weights distribution increases its mean and coefficient

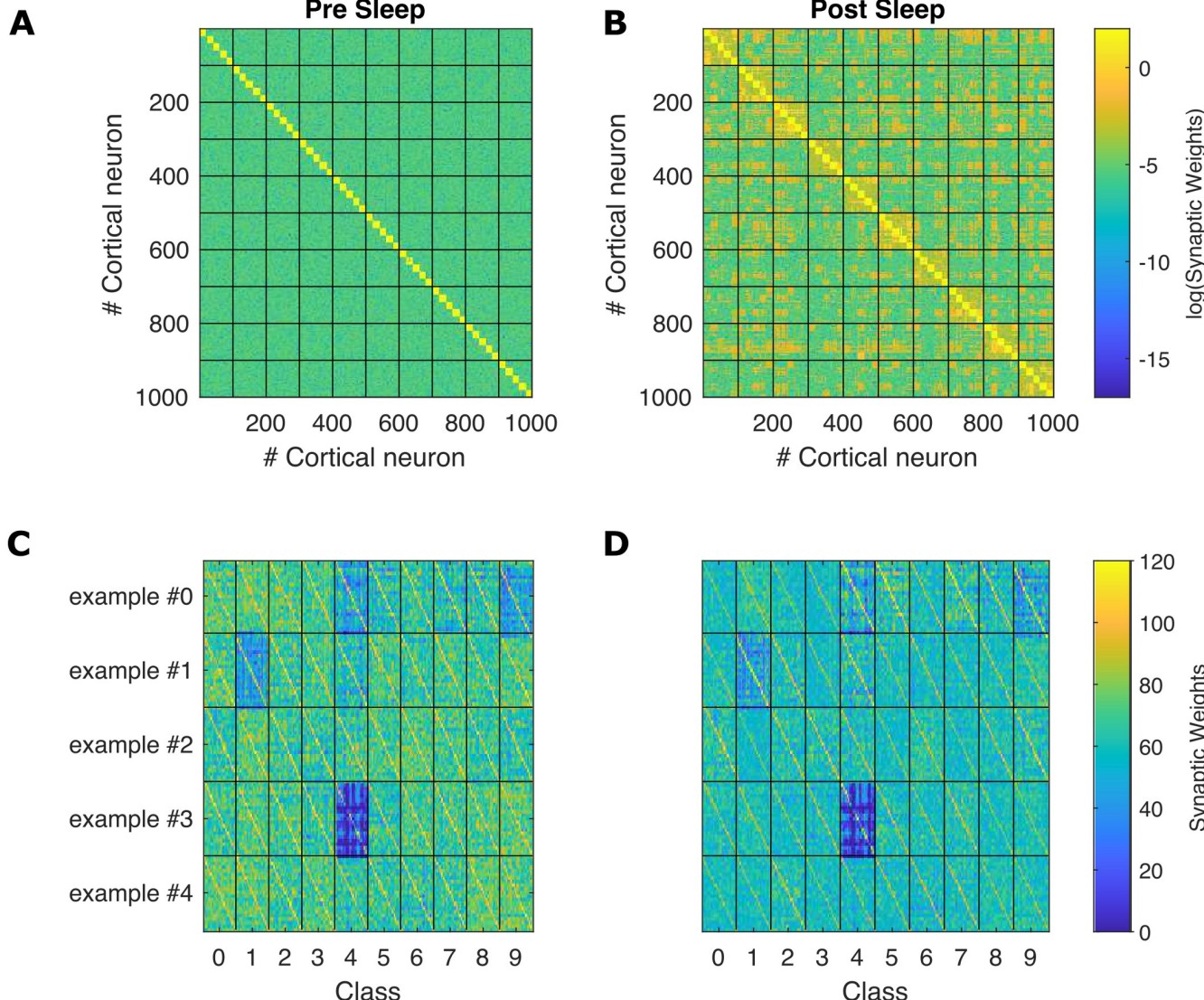

**Fig 6. Effects of sleep on intra-class and example-specific synapses after training with a noisy contextual signal.** Comparison of synaptic-weight matrices, pre-sleep (**Left**) vs post-sleep (**Right**). Training over 5 examples per class (20 neurons per example). **A)** and **B)** depict all cortico-cortical synaptic weights connecting the full set of trained neurons (colour bar, logarithm scale), black lines separate neurons solicited by contextual signal together with thalamic sensorial signal pointing to images belonging to the same digit class in the training phase; **C)** and **D)** focus on the synaptic weights connecting groups of cortical neurons simultaneously solicited by a contextual signal in the training phase (thus stimulated over the same sensorial signal identifying images belonging to the same class) (colour bar, linear scale), vertical black lines separate neurons trained over different categories, horizontal black lines separate cortical groups of neurons solicited during the presentation of the same 10 examples (one per digit class). The post-sleep intra-class cooperation is evident in B and in agreement with Fig 5B.b, while the homeostatic effect over example-specific synapses is manifest in D as already suggested by Fig 5B.a. The strong, example-specific differences in synaptic weights (e.g. *third example of class* 4) are due to the noisy training protocol that introduces randomness in the magnitude of the contextual signal that reaches the example-specific group.

of variation (from $\mu = 0.005$, $\mu/\sigma = 0.001$, skewness 0.003 pre-sleep to $\mu = 0.5$, $\mu/\sigma = 1.6$, skewness $-2.9$ post-sleep).

Specifically, the homeostatic effect of deep-sleep-like SOs can be identified comparing Fig 6C and 6D: the distribution of synaptic weights sharpens (i.e. it exhibits smaller post-sleep $\sigma$ and $\mu$) and presents a general depression of synaptic weights. These two variations combine to produce beneficial effects. First, this leads to a lowering of the heterogeneity of representation of learned examples, a reduction of the energetic cost of memory recall (reduced synaptic strength is associated with a lower metabolic cost of synaptic activity) and lower post-sleep spiking rates (see section 2.1). Moreover, deep-sleep-like oscillations affects categorical association and is depicted in Fig 5B.b: synapse weights connecting groups of neurons trained over different examples but belonging to the same digit class increase from a nearly zero pre-sleep value, while synapses connecting representations of memories belonging to different classes are much less affected. This effect is also visible comparing Fig 6A and 6B, where synapses connecting representations of memories belonging to the same digit class light up (big squares along the diagonal). Asymmetric STDP induces, on one hand, the depression of strong synapses, on the other the association among neuronal groups coding for the same class (i.e. trained over similar stimuli), through a mechanism of resemblance in their thalamic representation.

## 3 Discussion

We propose a simplified thalamo-cortical spiking model (ThaCo) that exploits the combination of context and perception to build a soft-WTA circuit, and that is able to express sleep-like slow oscillations. In order to be compliant with biological rhythms, we first verified that the proposed network is able to reproduce the experimental measures of neuronal firing rates during awake and deep-sleep states performed by Watson et al [25]. The agreement with the experiments has been achieved by further developing the thalamo-cortical spiking model proposed in [27] and by setting the model parameters to better fit the experimental recordings. The model we propose is capable of fast incremental learning from few examples (its performances are comparable to those expressed by Knn, of rank increasing with the number of examples) and of alternating several learning-sleep phases; moreover, it demonstrates resilience when subjected to noisy perceptions with better performances than Knn algorithms; these three facts constitute significant extensions to the previous study [27].

In recent years, there has been growing interest in the development of artificial neural networks (ANNs) or deep neural networks inspired by features found in biology, yet still using mechanisms for learning and inference which are fundamentally different from what is actually observed in biology. On the other hand, there is also a plenty of computational models aiming at reproducing biological proprieties in an exact way. Many models have been proposed for pattern recognition tasks that use biologically-plausible mechanisms, combining spiking networks and STDP plasticity [41–43]. The ThaCo model has been developed in line with this philosophy, delivering a spiking neural network which relies on a combination of biologically plausible mechanisms. It uses conductance-based AdEx neurons, STDP and lateral inhibition. A crucial ingredient, which mostly differentiates our approach from previous works, is the introduction of a contextual signal which drives the training procedure, making it similar to a target-based approach [44, 45] and enabling huge advantages in terms of training velocity and precision. Such mechanism was inspired by the work done by Larkum [1] suggesting that the activity of a neuron is amplified when it receives a coincidence of signals from both lower and higher levels of abstraction. This allows the recruitment of new neurons to learn novel examples through the incremental building of a soft-WTA mechanism.

We stress that, even though we showed to be successful in reproducing specific experimental observations, the aim of this work is not to exactly reproduce a biological network (for instance, the emulation of metabolic processes goes beyond the scope of this work), but to develop a simplified task-specific spiking neural network able to express biological features, and receive an indication on how even an approximated emulation of deep-sleep and of the combination of contextual and perceptual information can positively affect the network performances. Specifically, without any pretence to be biologically realistic, the neuron model used in these simulations is a point-like AdEx (see section The neuron model), yet we are able to emulate a compartment neuron behaviour (described in [1]) without introducing more complex morphological units in the network. For this, we approximated the coincidence mechanism by setting both the contextual and sensory inputs impinging on cortical neurons in a subthreshold point (see 4.1).

Another major aspect of our work is the effect of sleep on the network and on the memories stored in it. The role played by sleep in memory consolidation has been widely studied from an experimental point of view [46, 47], but only recently it has become the object of theoretical and computational modelizations [27, 48–50]. In our work, we investigated computationally the effect of slow oscillations on the structure and the performances of the network when the STDP plasticity is turned on. We proved that deep-sleep-like slow oscillations can be beneficial to equalize the memories stored in a cortico-thalamic structure when learned in noisy conditions. Indeed, slow oscillations can compensate for the contextual noise through homeostasis, equalizing synaptic weights and creating beneficial associations that improve classification performance.

The predictions of our model are also a first step toward the reconciliation of recent experimental observations about both an average synaptic down-scaling effect (synaptic homeostatic hypothesis—SHY [51]) and a differential modulation of firing rates [25] induced by deep-sleep, which is believed to be a default state mode for the cortex [52].

As mentioned above, we focused on the role of NREM-sleep for memory consolidation. The simulation of a complete sleep cycle that includes REM and micro-arousal phases goes beyond the scope of this paper and is currently under investigation. One more limitation of this work is that it does not take into account the role of synchronization among different brain regions. Actually, assuming a typical neural density in the range of $5 \cdot 10^4$ neurons per $mm^2$ of the cortex, and considering that the maximum size of the proposed model rises up to 5000 cortical neurons, such a number is equivalent to a small cortical area with a dimension of about $300 \mu m$, that is well below the size of a single cortical area. To overcome this limitation, we are extending the model to multi-layers and multi-area descriptions.

Finally, this work represents an additional contribution in understanding sleep mechanisms and functions, in line with the efforts we are carrying out in data analysis [53, 54] and in large-scale simulations [55], aimed at bridging different elements in a multi-disciplinar approach. In particular, it hints to a careful balance between architectural abstraction and experimental observations as a valid methodology for the description of brain mechanisms and of their links with cognitive functions.

## 4 Materials and methods

The results of the ThaCo model (Section 2) have been obtained thanks to fine implementations of several features. Such fine-tuning is presented in this Section and in the S1 Text. In particular, Section 4.1 addresses the crucial point of the model calibration, aimed at inducing a soft-WTA mechanism by combining context and perception, achieved by setting the network parameters in what we call an *under-threshold regime*, that enables a training through the

selected STDP model on single-compartment standard Adaptive Exponential integrate-and-fire neuron (AdEx) that would not otherwise distinguish among basal and apical stimuli (see S1 Text). MNIST characters are coded by the thalamus according to the scheme presented in S1 Text that preserves a notion of distance among visual features.

Simulation reported were executed on dual-socket servers with eight-core Intel(R) Xeon(R) E5–2620 v4 CPU per socket. The cores are clocked at 2.10GHz with HyperThreading enabled, so that each core can run 2 processes, for a total of 32 processes per server. The ThaCo model has been implemented using the NEST 2.12.0 [56] simulation engine.

## 4.1 Winner-take-all mechanisms by combining context and perception

We set the network parameters to induce the creation of WTA mechanisms by emulating the organizing principle of the cortex described by Larkum et al. [1].

During the training, the network is set in a hard-WTA regime (firing rate different from zero only on a selected example-specific sub-set of neurons), while during classification it works in a soft-WTA regime (*i.e.* the firing rate can be different from zero in multiple groups of neurons, and the winner group is assumed to be the one firing at the higher rate). Specifically, during the training, we set our parameters to be so that the thalamic signal alone is not sufficient to make neurons spike. This is reported in Fig 7C that represents the mean firing rate and the membrane potential over time for a group of cortical neurons stimulated to encode for a training example in three different contextual scenarios: Fig 7C-center shows the network behaviour in the absence of a thalamic signal; Fig 7C-left shows the network response without the contextual signal; Fig 7C-right shows the network behaviour with both the contextual and the thalamic signal. The cortical activity in the absence of contextual signal is null and it is really low when only the stimulation of the contextual signal is present. The combined action of the two, on the other hand, yields a higher spiking activity. We can therefore conclude that we put the network in what we named an *under-threshold regime*. Moreover, to better show the implemented soft Winner-take-all dynamics, we present the mean firing rate of three subgroups of cortical neurons trained over different examples belonging to different categories during both retrieval (i.e. training examples are presented again to the network without any contextual signal) and classification phases. The implementation of WTA dynamics is depicted in Fig 7D.

**4.1.1 Simple mathematical model of soft-WTA creation.** In this section, we discuss the capability of our model to learn over a few examples through soft-WTA mechanisms. First, we demonstrate how the network is surely endowed with the capability to behave like a Knn classifier. In the first training step, the network is exposed to one example for each of ten digit class ($L = 10$). Let $D^{(l)} = \{1 + (l-1)K, \ldots, lK\}$ be the set of indices of the $K$ excitatory cortical neurons that are induced to fire by the simultaneous presence of the contextual stimulation and the thalamic input, carried by $T$ thalamic neurons (see Fig 7C) when presented with one of the training examples $l \in \{1, .., L\}$. Also, starting from an initial value $w_0^{th \rightarrow cx}$, let $w_{eq}^{th \rightarrow cx}$ be the final average weight induced by STDP on the connections between the thalamic excitatory neurons that are active during the learning of the training example $l$ and the $K$ excitatory cortical neurons that are induced to activity. Finally, let $\mathbf{x}_{th}^{(l)}$ be the binary feature vector of the training example $l$. The average weight at equilibrium of the connections between the thalamic neurons activated by the example $l$ and the excitatory cortical neurons can be written as:

$$w_{nj}^{th \rightarrow cx} = (w_{eq}^{th \rightarrow cx} - w_0^{th \rightarrow cx})x_{th,j}^{(l)} + w_0^{th \rightarrow cx} \quad j \in \{1, .., T\},$$
$$n \in D^{(l)} = \{1 + (l-1)K, .., lK\}, \quad l \in \{1, \ldots, L\}$$

(1)

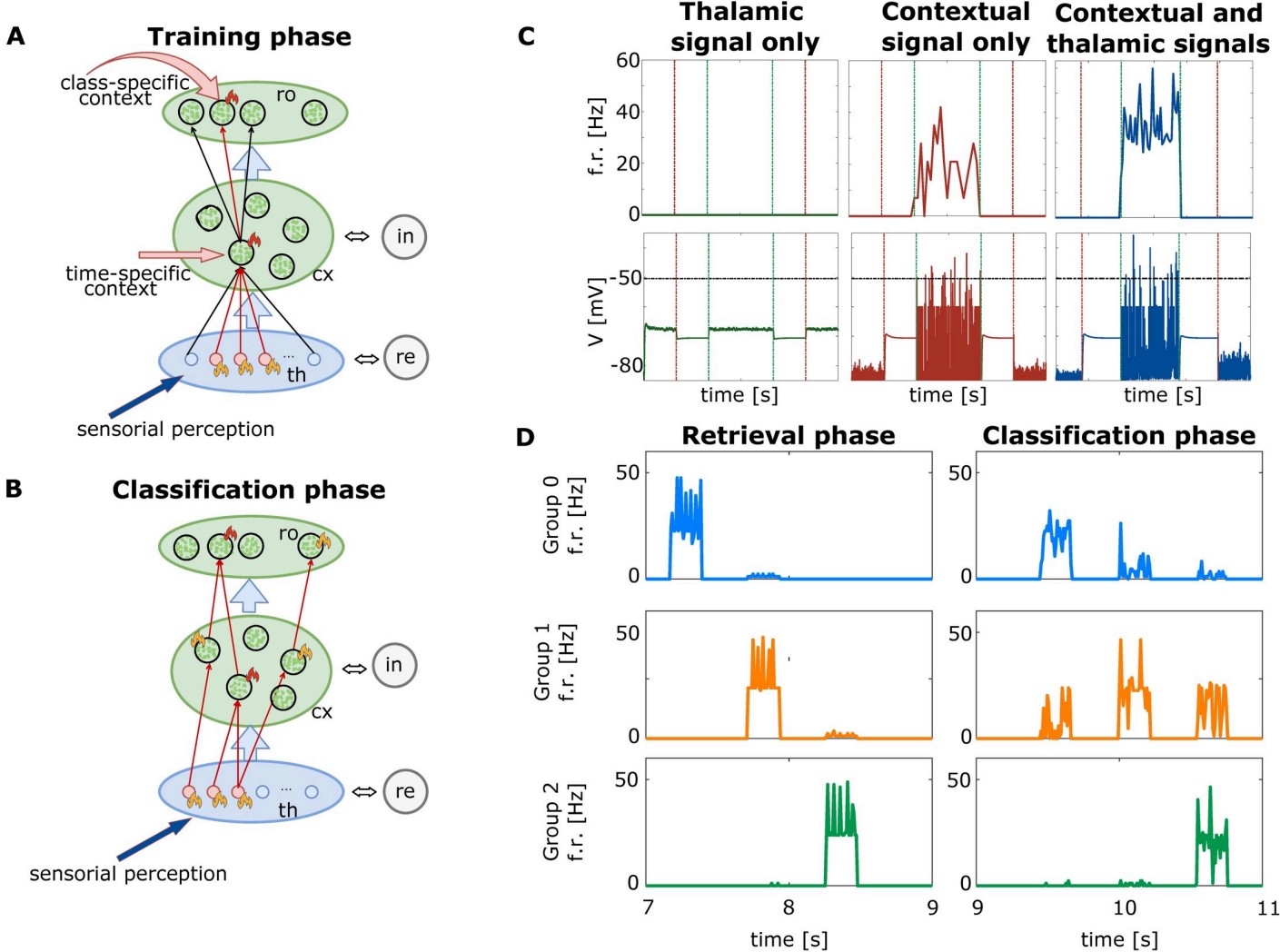

**Fig 7. Training and classification phases. A)** Training Phase: the *sensorial perception* (blue arrow) is encoded into the thalamic excitatory neurons. A *time-specific contextual signal* is delivered to a subset of cortical excitatory neurons (red arrow), raising their perceptual threshold on the example-specific thalamic activity and inducing in such group of neurons a high firing rate during training (represented with red flames drawing). STDP induces group-specific connectivity in the subset of facilitated cortical neurons, and the thalamic pattern is sculptured into synapses that connect the thalamus with the example-specific group. The readout layer, made of as many groups of cortical neurons as the number of classification classes, is also trained through the simultaneous administration of a *class-specific contextual signal* addressing the subset associated to the correct class label (red arrow). **B)** Classification Phase: no contextual signal is given to the cortex. One or more subgroups of excitatory cortical neurons reach a high (red flames), intermediate (yellow flames) or low (no flames) level of activity, depending on the similarity between the stimulating thalamic pattern and the training set. Here, the WTA mechanism is essential to decide the classification answer; the network decision can be evaluated measuring the activation level of the groups, either in the *cortex* or in the *readout* layer. **C)** Combination of contextual and perceptual signals to create one group of cortical neurons sensible to a specific example in a soft winner-take-all mechanism. Three example are presented to three cortical groups for 2*s* (start and stop presentation time marked by green and red dashed lines). High firing rate is induced only when the example-specific cortical group is reached by both the thalamic (perceptual) stimulus and the contextual (example-specific) signal. Upper row: mean firing rates of a cortical subgroup of cortical neurons; lower row: mean membrane potentials (the black dotted line depicts the firing threshold potential). Left column: Neuron activity when stimulated by the thalamic signal only (perception): null firing rate and under-threshold membrane potential. Central column: Neuron activity when stimulated by contextual signal only (internal prediction): a moderate firing rate is induced. Right column: Simultaneous perceptual and contextual signals induce a high firing rate in the example-specific group. **D)** Soft Winner-take-all dynamics among example-specific groups of cortical neurons during retrieval and classification phases. Mean firing rates of three groups trained to be sensitive to three different examples. **D-Rows)** Firing rates of the firs (blue), second(orange), third (green) neural group. **D-Retrieval phase column)** Exactly the three learnt examples (belonging to three different digit classes) are re-presented to the network without any contextual signal, resulting in an almost hard-WTA dynamics among the three groups. **D-Classification phase column)** Three novel images, for which the network has not been trained, are presented to the network without any hint from the contextual signal; a soft-WTA dynamics is emerging, rewarding for each presentation the neuron group with the highest firing rate and still allowing all the other groups to fire with non-zero probability. Here read-out neurons are not represented and the figure demonstrates how it is possible to extract a classification answer looking at the cortical layer only.

After the training on the first set of $L$ examples, a total of $C = kL$ cortical neurons will have been exposed to the combination of contextual and thalamic stimulation (see Fig 3A). During the classification phase, represented in Fig 3B, when a never seen stimulus (the image $S$ to be classified) is presented to the network, the average signal from the thalamic layer (composed of $T$ neurons) to the excitatory cortical neurons (in all the $L$ trained cortical groups) is:

$$\bar{g}_{n,th\to cx}^{(S)} = \sum_{j=1,T} w_{nj}^{th\to cx} \rho_{th} x_{th,j}^{(S)} \qquad j \in \{1,..,T\} \quad n \in D^{(l)}, \quad l \in \{1,\ldots,L\} \tag{2}$$

where $\rho_{th}$ is the rate of the active thalamic neurons, and $\mathbf{x}_{th}^{(S)}$ is the binary thalamic feature vector of the novel image $S$ to be classified. Assuming $w_0$ to be much smaller than $w_{eq}$, the average signal from thalamic neurons to each cortical neuron belonging to $D^{(l)}$ group can thus be written as:

$$\bar{g}_{n,th\to cx}^{(S)} \simeq \sum_{j=1,T} w_{eq}^{th\to cx} \rho_{th} x_{th,j}^{(l)} x_{th,j}^{(S)} = w_{eq}^{th\to cx} \rho_{th} \mathbf{x}_{th}^{(l)} \cdot \mathbf{x}_{th}^{(S)} \qquad \text{for } n \in D^{(l)} \tag{3}$$

where $S$ is the novel stimulus presented during the classification phase and $l$ is the learning example over which the set of neurons $D^{(l)}$ have been trained. The vectors of thalamic features can be normalized ($\mathbf{u} = \mathbf{x}/N(\mathbf{x})$, using their Euclidean norm ($N(\mathbf{x}) = \|\mathbf{x}\|_2 = (\sum_i x_i^2)^{\frac{1}{2}}$). The Euclidean distance among each training example ($l$) and the images ($S$) to be classified can be written as $d_{l,S} = \|\mathbf{u}^{(l)} - \mathbf{u}^{(S)}\|_2$. It follows that $d_{l,S}^2 = 2 - 2\mathbf{u}^{(l)} \cdot \mathbf{u}^{(S)}$, where we used the normalization condition for both $\mathbf{u}^{(l)}$ and $\mathbf{u}^{(S)}$. In this way Eq 3 can be rewritten as:

$$\bar{g}_{(l),th\to cx}^{(S)} \simeq w_{eq}^{th\to cx} \rho_{th} N^{(l)} N^{(S)} \mathbf{u}^{(l)} \cdot \mathbf{u}^{(S)} = w_{eq}^{th\to cx} \rho_{th} N^{(l)} N^{(S)} \left(1 - \frac{1}{2} d_{l,S}^2\right) \tag{4}$$

Eq 4 tells us that the thalamic signal is a decreasing function of the distance $d_{l,S}$, if all training examples are equally normalized($\|\mathbf{x}^{(i)}\|_2 = \|\mathbf{x}^{(j)}\|_2 \forall i, j \in 1, ..L$) and neglecting for a while the possible changes in the thalamic rate $\rho_{th}$, that in our model can be mediated by the existing cortico-thalamic feedback path. Under the approximation of constant $\rho_{th}$ we can immediately show that, after having being exposed to the first set of training examples, the soft-WTA ThaCo excitatory network is at least endowed with the capability to behave as a nearest neighbour classifier of the first order (Knn-1 classifier). The winning candidate $K$ among the $L$ competing cortical groups is initially suggested to the network as the one reached by the strongest thalamic stimulus when presented with the never seen image $S$:

$$\text{initial candidate } K(S) = \arg_l \max[\bar{g}_{(l),th\to cx}^{(S)}] = \arg_l \min[d_{l,S}] \qquad \text{for } l \in 1, .., L \tag{5}$$

Indeed, under the assumption that the neuron activity depends on the incoming signal (both excitatory and inhibitory) through a transfer function $\mathcal{F}(g)$ monotonically increasing over the total incoming current $g$, we will now show that: 1) the role of inhibition will be to help the computation of (a soft) argmax; 2) the recurrent intra-group cortical excitation provides an additional boost to the selection of the winner. To confirm this, we shall now consider explicitly the contribution of both recurrent and inhibitory contributions. The total average input signal to each cortical neuron depends on the group $l$ the neuron belongs to and on the the stimulus $S$ to be classified:

$$\bar{g}_{(l),tot}^{(S)} = \bar{g}_{(l),th\to cx}^{(S)} + \bar{g}_{cx\to cx(l)}^{(S)} + \bar{g}_{inh\to cx(l)} \tag{6}$$

Under the approximation of constant $\rho_{th}$, the first term in Eq 6 is provided by Eq 3.

Concerning the second term, the training protocol illustrated by Fig 7 creates cortico-cortical synapses of strength $w_{eq}^{cx \to cx}$ only among neurons belonging to the same group $l$, *i.e* among neurons trained on the same example, while connections among neurons selective for different training examples are left to the initial value $w_0^{cx \to cx}$. Assuming that $w_0^{cx \to cx} \ll w_{eq}^{cx \to cx}$, after learning we have:

$$w_{l_1,l_2}^{cx \to cx} = w_{eq}^{cx \to cx} \delta_{l_1,l_2} + w_0^{cx \to cx}(1 - \delta_{l_1,l_2}) \sim w_{eq}^{cx \to cx} \delta_{l_1,l_2} \qquad l_1, l_2 \in 1, .., L \qquad (7)$$

Under the assumption that the activities of the $K$ neurons belonging to the same subgroup ($l$) are similar to each other, the second term in Eq 6 reduces to the recurrent intra-group excitatory contribution:

$$\bar{g}_{cx \to cx(l)}^{(S)} \sim \bar{g}_{cx(l) \to cx(l)}^{(S)} = (k - 1) \cdot w_{eq}^{cx \to cx} \bar{\rho}_{(l)}^{(S)} \qquad (8)$$

where $\bar{\rho}_{(l)}^{(S)}$ is the average firing rate reached by the cortical group $l$ when activated by the novel stimulus $S$.

In our simplified mode, all $w^{cx \to inh}$ and $w^{inh \to cx}$ synapses are non-plastic and set to an identical value. Therefore, the third term, the input signal from cortico-cortical inhibition, is in our architecture equal to:

$$\bar{g}_{inh \to cx(l)}^{(S)} = \bar{g}_{inh \to cx}^{(S)} = N_{inh} \cdot w^{inh \to cx} \bar{\rho}_{inh}^{(S)} \qquad (9)$$

where $N_{inh}$ is the number of cortical inhibitory neurons and $\rho_{inh}$ is the inhibitory neurons activity.

In summary, Eq 6, *i.e.* the total current stimulating each of the $L$ groups of cortical neurons responding to the thalamic stimulus $S$ can be reformulated:

$$
\begin{aligned}
\bar{g}_{(l),tot}^{(S)} &\sim \bar{g}_{(l),th \to cx}^{(S)} + \bar{g}_{cx(l) \to cx(l)}^{(S)} + \bar{g}_{inh \to cx}^{(S)} = \\
&= w_{eq}^{th \to cx} \rho_{th} N^{(l)} N^{(S)} \left(1 - \frac{1}{2} d_{l,S}^2\right) + (k - 1) \cdot w_{eq}^{cx \to cx} \bar{\rho}_{(l)}^{(S)} + N_{inh} \cdot w^{inh \to cx} \bar{\rho}_{inh}^{(S)}
\end{aligned}
\qquad (10)
$$

When the average rate is well below saturation, its relationship to the total input signal is well described by a threshold-linear function:

$$\bar{\rho}_{(l)}^{(S)} = \alpha(\bar{g}_{(l),tot}^{(S)} - g_{\mathrm{thresh}}) H(\bar{g}_{(l),tot}^{(S)} - g_{\mathrm{thresh}}) \qquad (11)$$

where $\alpha$ is a constant coefficient, $H$ is the Heaviside function and $g_{\mathrm{thresh}}$ is the firing threshold. Therefore, assuming that the input signal is above threshold ($\bar{g}_{(l),tot}^{(S)} > g_{\mathrm{thresh}}$), we have that

$$\bar{g}_{(l),tot}^{(S)} = \frac{w_{eq}^{th \to cx} \rho_{th} N^{(l)} N^{(S)} \left(1 - \frac{1}{2} d_{l,S}^2\right) + N_{inh} \cdot w^{inh \to cx} \bar{\rho}_{inh}^{(S)} - \alpha(k - 1) \cdot w_{eq}^{cx \to cx} g_{\mathrm{thresh}}}{1 - \alpha(k - 1) \cdot w_{eq}^{cx \to cx}} \qquad (12)$$

we also require that $\alpha(k - 1) \cdot w_{eq}^{cx \to cx} < 1$, i.e. self feedback should be smaller than one, otherwise the system would become unstable.

Considering that the inhibitory signal is equal for all $L$ groups under the provisional assumption of constant $\rho_{th}$, i.e. no cortico-thalamic feedback), Eq 12 tells us that the final choice of the network would confirm the initial guess of Eq 5:

$$\text{winning } K(S) = \arg_l \max[\bar{g}_{(l),tot}^{(S)}] = \arg_l \min[d_{l,S}] \quad l \in 1, .., L \qquad (13)$$

*i.e.* the network tends to a stationary condition in which the $L$ groups of $K$ neurons can be set at different firing rates that decrease with the distance $d_{l,S}$. Moreover, the readout layer

combines signal coming from groups of cortical neurons trained over different examples yet belonging to the same class: thus, the network expresses a behaviour similar to that of a a higher-order Knn—n,.

## 5 Supporting information

**S1 Text. Mechanisms and implementation details.** The reader can find in this section details about the mechanisms in action in ThaCo during the Training and Classification phases (see the 'Training and classification phases' and 'Balanced mini-batch training' paragraphs, specifically Fig A providing a comparison of incremental learning protocols) and details concerning the implementation and effects of deep sleep dynamics (see 'Sleep-like oscillatory dynamics'). The specific form of STDP plasticity is described in paragraph 'Spike-Timing-Dependent Plasticity' (and depicted in Fig B showing the effects of deep-sleep on network performances) and the neuronal model in 'The neuron model'. Details about the handwritten digits datatsets are presented in the 'The datasets of handwritten characters' paragraph, while paragraph 'Thalamic coding of visual stimuli' describes the fuzzy-logic-inspired pre-processing algorithm, adopted for a tuning of thalamic activity that preserves a notion of distance among visual features. The 'Salt-and-pepper noise' paragraph is about the method used to add noise to images during training and classification. Finally, the set of parameters needed to configure the spiking model is provided in Table A, paragraph 'Parameters of the spiking model'.
(PDF)

## Acknowledgments

We thank the artist Lorenzo PONT Pontani for cat drawing in Fig 1.

## Author Contributions

**Conceptualization:** Bruno Golosio, Chiara De Luca, Cristiano Capone, Elena Pastorelli, Pier Stanislao Paolucci.

**Data curation:** Chiara De Luca, Elena Pastorelli.

**Formal analysis:** Bruno Golosio, Chiara De Luca, Cristiano Capone, Giovanni Stegel, Giulia De Bonis, Pier Stanislao Paolucci.

**Funding acquisition:** Pier Stanislao Paolucci.

**Investigation:** Bruno Golosio, Chiara De Luca, Cristiano Capone, Elena Pastorelli, Gianmarco Tiddia, Pier Stanislao Paolucci.

**Methodology:** Bruno Golosio, Chiara De Luca, Cristiano Capone, Giovanni Stegel, Pier Stanislao Paolucci.

**Project administration:** Pier Stanislao Paolucci.

**Resources:** Chiara De Luca, Elena Pastorelli.

**Software:** Bruno Golosio, Chiara De Luca, Cristiano Capone, Elena Pastorelli, Gianmarco Tiddia.

**Supervision:** Chiara De Luca, Cristiano Capone, Elena Pastorelli, Giulia De Bonis, Pier Stanislao Paolucci.

**Validation:** Chiara De Luca, Cristiano Capone, Elena Pastorelli.

**Visualization:** Chiara De Luca, Cristiano Capone.

**Writing – original draft:** Bruno Golosio, Chiara De Luca, Cristiano Capone, Elena Pastorelli, Pier Stanislao Paolucci.

**Writing – review & editing:** Bruno Golosio, Cristiano Capone, Elena Pastorelli, Giulia De Bonis, Pier Stanislao Paolucci.

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
