## [Decision Letter · Decision Letter 0]

11 Jan 2021

Dear Mss. De Luca,

Thank you very much for submitting your manuscript "Thalamo-cortical spiking model of incremental learning combining perception, context and NREM-sleep-mediated noise-resilience" for consideration at PLOS Computational Biology.

As with all papers reviewed by the journal, your manuscript was reviewed by members of the editorial board and by several independent reviewers. In light of the reviews (below this email), we would like to invite the resubmission of a significantly-revised version that takes into account the reviewers' comments.

We cannot make any decision about publication until we have seen the revised manuscript and your response to the reviewers' comments. Your revised manuscript is also likely to be sent to reviewers for further evaluation.

Sincerely,

Maxim Bazhenov

Guest Editor

PLOS Computational Biology

Samuel Gershman

Deputy Editor

PLOS Computational Biology

Reviewer's Responses to Questions

**Comments to the Authors:**

Reviewer #1: In this study, the authors use a spiking model of thalamocortical network that is capable of learning to examine the role of sleep like state. Using the oscillatory input that induced activity in the thalamocoritcal network for sleep state, the study finds that it improve subsequent learning. Overall, the study identifies interesting results, however requires several issue to be addressed.

- The description of the sleep state is not presented in the manuscript. The author's refer to their previous publication in the main text and include very few details about sleep in the methods section. Please include additional details about how sleep is implemented in both main text and also in the methods.

- Based on author's previous paper, I assume sleep involves oscillatory input applied to cortical neurons. How does the frequency and duration of the oscillation impact the results ?

- Another critical issue in the manuscript is about what is referred as incremental learning. In this manuscript, incremental learning refers to increasing the number of training set across all classes. However, typically the term incremental learning (often used in machine learning) involving tasks such as MNIST involve learning different classes incrementally. While this distinction may be subtle, it is critical since learning classes is known to lead to forgetting.

- The authors suggest that homeostatic effect and sharpening of the synaptic weight is critical for the beneficial effect of sleep. It would be helpful to try manipulation of the homeostatic effect and sharpening by changing the STDP rules, and examining the results. This way, the postulated mechanisms could be verified.

- It is not clear what is the role of inhibitory neurons and TC and RE cells in thalamus during sleep ?. Biological experiments, suggest that inhibitory and thalamic neurons play important role in synchronization across regions, and is helpful to examine their role in this network.

Reviewer #2: I applaud the efforts of the authors to create a model inclusive of cortical structure, thalamocortical interactions, STDP, sleep homeostasis and other elements. The paper appears to have some interesting observations and appears to be based on a great deal of work modifying and old model. However, many crucial details appear not to be explained, rendering it impossible for me, as a biologist, to understand the basic experiments done here. Those need to be clarified before I can fully evaluate this paper.

MAJOR ISSUES:

How was the spike rate homogenization shown in Figure 2 instantiated? Why did this occur? Did it naturally fall out of the network with no intentional design? Was it explicitly added to re-create the biological situation? I find a great deal is lacking in the description of how this was carried out.

It would be logical to test whether the log-normal distribution of neuronal firing rates was crucial to the network results here. That appears to have been speicifically added here, but it was not shown whether it was important

Similarly, the effect of homogenization during sleep was added specially here, but is not investigated

I’d like to learn more about the sleep used. This is not clearly described, although delta waves are mentioned. Are there UP and DOWN states - or essentially bursts of firing between delta waves? What firing dynamics are assumed in these bursts of firing? How might those work via STDP to achieve the synaptic downscaling shown? Or was it just the homogenization of firing rates shown in figure 2? Some of what was done may be reminiscent of Levenstein et al 2017: https://www.ncbi.nlm.nih.gov/pmc/articles/PMC5511069/, but I cannot tell if that is what is used here. Or really any details about the sleep paradigm. These need to be explained more clearly if the reader is to understand the experiments done.

I think the basic goals, logic and setup of the training and classification need to be explained. It is not directly explained anywhere, but after much reading and confusion, I was able to understand that digit stimuli are given to the network and then classification must be a test to determine whether new stimuli are placed into the proper number category based on the criteria described around line 192. This, or whatever logic should be clearly explained. Also it should be clearly explained that multiple stimuli are given during training, but only a single stimulus is shown during classification.

Is accuracy based on the network classifying the number stimuli into the correct (ie human-based) number categories? Again, I don’ think I saw any of this explained and it took me a long time to piece it together.

I gather that my lack of understanding of this is based on my background in biology rather than ANNs, but explaining this basic logic will help many readers and is necessary for my review.

I still do not understand the experimental set-up implied in the “synapses trained over the same example” and other elements of figure 5B. I am afraid I still do not understand the fundamentals of the work here, since they were not explicitly explained. I was surprised to see that perhaps not all neurons in the network exposed to all examples. Am I correct in understanding that? If not, can the authors better explain the training paradigm and how the different groups in 5B are generated. I think training needs to be fully explained, as does classification measurement.

In Figure 6, how does a “neuron belong to a different category” (as stated in Fig 6 legend)? Does that mean it responds maximally to a different stimulus? Or was trained on only one stimulus? Again, the details of training are not explained, making the paper difficult to understand.

Figures S1 should be brought early into the paper as part of an explanation of the basic experiment as mentioned above.

MINOR ISSUES:

I cannot tell if “ro” as in figure 1 is defined anywhere. I assume it stand for “readout”

Line 62; “higher rate” maybe should be “highest rate”

Line 136: “mice” should be “rats”

**Have all data underlying the figures and results presented in the manuscript been provided?**

Reviewer #1: Yes

Reviewer #2: Yes

PLOS authors have the option to publish the peer review history of their article (what does this mean?). If published, this will include your full peer review and any attached files.

Reviewer #1: No

Reviewer #2: No
---

## [Decision Letter · Decision Letter 1]

5 May 2021

Dear Mss. De Luca,

We are pleased to inform you that your manuscript 'Thalamo-cortical spiking model of incremental learning combining perception, context and NREM-sleep-mediated noise-resilience' has been provisionally accepted for publication in PLOS Computational Biology.

Best regards,

Samuel J. Gershman

Deputy Editor

PLOS Computational Biology

Reviewer's Responses to Questions

**Comments to the Authors:**

Reviewer #1: The authors addressed the main issues raised in the updated manuscript.

Reviewer #2: The paper is greatly clarified now. Thank you.

**Have the authors made all data and (if applicable) computational code underlying the findings in their manuscript fully available?**

Reviewer #1: None

Reviewer #2: Yes

PLOS authors have the option to publish the peer review history of their article (what does this mean?). If published, this will include your full peer review and any attached files.

Reviewer #1: **Yes: **Giri Krishnan

Reviewer #2: No

---

## [Editor Report · Acceptance letter]

14 Jun 2021

PCOMPBIOL-D-20-00684R1 

Thalamo-cortical spiking model of incremental learning combining perception, context and NREM-sleep

Dear Dr De Luca,

I am pleased to inform you that your manuscript has been formally accepted for publication in PLOS Computational Biology. Your manuscript is now with our production department and you will be notified of the publication date in due course.

With kind regards,

Zsofi Zombor
